# Gait kinetics before and after total hip arthroplasty in people with unilateral hip osteoarthritis

Lingling Zhong[1], Patrick Wai-Hang Kwong[1*], Jack Jiaqi Zhang[1], Ananda Sidarta[2], Clare Chung-Wah Yu[1]

1 Department of Rehabilitation Sciences, The Hong Kong Polytechnic University, Kowloon, Hong Kong, China, 2 Rehabilitation Research Institute of Singapore, Nanyang Technological University, Singapore, Singapore

* wai-hang.kwong@polyu.edu.hk

## Abstract

### Background

Total hip arthroplasty (THA) is a common intervention for end-stage osteoarthritis (OA) that improves gait kinetics. However, full restoration of mobility through THA remains elusive. Limited studies have examined changes in hip kinetics throughout the entire stance phase. In this study, we explored the differences in hip moment and power between preoperative and postoperative states in unilateral hip OA and compared these patients with healthy controls.

### Methods

A secondary analysis was conducted using a publicly available dataset. A total of 69 healthy controls and 67 participants with a history of THA for whom preoperative and postoperative data were available for analysis were included in the study. Motion capture data obtained using the Plug-in Gait marker set was analyzed and modeled in Visual3D. Statistical parametric mapping (SPM) paired $t$ tests were used to determine the differences in hip moment and power during the stance phase between preoperative and postoperative states. Independent-samples $t$ tests were conducted to compare these metrics in healthy controls and both preoperative and postoperative groups. SPM regression was used to analyse the correlation between changes in walking speed and hip kinetics.

### Results

Significant changes in hip sagittal moment (0.0–4.3% stance phase, $P=0.037$; 66.7–100.0% stance phase, $P<0.001$), frontal moment (9.9–38.7% stance phase, $P<0.001$; 55.2–93.7% stance phase, $P<0.001$) and hip power (50.8–70.8% stance phase, $P<0.001$; 84.2–100.0% stance phase, $P<0.001$) were observed between the

**Data availability statement:** The data are available in public repository (https://www.nature.com/articles/s41597-022-01483-3).

**Funding:** This research was funded by the Hong Kong Polytechnic University, grant number P0036617. The funder had no role in study design, data collection and analysis, decision to publish, or preparation of the manuscript.

**Competing interests:** The authors have declared that no competing interests exist.

preoperative and postoperative participants, and a minor difference was noted in the total difference duration between the postoperative participants and healthy controls. The postoperative participants experienced hip kinetic deficits, and a significant association was observed between hip kinetics and changes in walking speed.

## Conclusion

The hip kinetics of patients gradually normalise by 6 months after THA. Specific exercise programmes may be required to improve the specific gait patterns deficits of patients undergoing THA.

---

### Introduction

Total hip arthroplasty (THA) is a widely accepted and highly effective orthopaedic procedure that is primarily used for treating end-stage osteoarthritis (OA). In 2018, a total of 124,251 THA procedures were performed in France, a figure that is expected to rapidly increase in the future [1]. Total hip arthroplasty generally improves spatiotemporal gait parameters [2] and yields favourable clinical outcomes, including improved Harris hip scores [3], activity levels [4], and Western Ontario and McMaster Universities Osteoarthritis Index scores [3–5]. Despite these benefits, THA does not completely restore gait kinematics and kinetics to healthy control levels, even after 2 years post-operation [6]. According to systematic reviews [7,8], compared with healthy individuals, patients with a history of THA experience a reduction in walking speed and stride length, a reduction in the range of motion (ROM) of the hip joint, and a low peak hip abduction moment. Stief et al. [6] reported that the second external hip adduction moment in the terminal stance phase was higher in postoperative patients than in healthy controls. In other words, even after surgery, patients who undergo THA still exhibit significant differences in gait patterns compared to the normal population [9]. Studies have shown that impaired postoperative walking function can lead to fatigue, subsequently limiting patients' ability to perform daily activities and reducing their quality of life [10]. Therefore, it is essential to prioritize gait assessment as a key functional activity following THA. Quantitative gait analysis is required to better understand gait mechanics, as these mechanisms may underlie the functional limitations observed in patients who have undergone the procedure [11].

One-dimensional statistical parametric mapping (SPM1d) has been used to analyse kinematic and kinetic data [12] during walking [13] and running [14]. In SPM1d, one-dimensional continuous trajectories that exhibit changes over time or space are used for assessment [15]. SPM1d has also been used to analyse the outcomes of different surgical approaches [16] and changes in hip kinematics in patients after THA [17]. Pincheira et al. [16] observed significant changes in the hip adduction angle at 11% to 43% of the gait cycle among two distinct surgical approach groups. Kaufmann et al. [17] reported significant differences in the preoperative hip and knee flexion angles, which were confirmed using SPM1d. After THA, these angles significantly improved and became similar to those of the control group.

Previous studies have examined changes in hip kinetics following surgery [18,19]. Queen et al. [18] reported that the hip power on the surgical side during walking increased at 1 year post-operation. Lalevée et al [19] reported a significant reduction in hip sagittal moment in one-year postoperative participants compared to healthy controls, whereas no significant decrease was observed in the hip frontal moment. However, they focused only on the percentage of peak power and did not provide insights into changes in hip kinetics across different phases. To the best of our knowledge, no study has investigated changes in joint kinetics across the whole stance phase in postoperative patients. Six months after THA represents a clinically important time-point when most patients have achieved formal gait rehabilitation programs and achieved substantial functional improvement [7,20]. Horstmann et al. found that although patients showed improvements in gait symmetry and joint range of motion by six months, their dynamic hip and knee motions remain significantly lower than those of healthy controls, and muscle activity patterns differ substantially [4]. While Casartelli et al. demonstrated that at six months post-THA, patients walk with velocity, cadence, and support times that are comparable to healthy individuals [21]. Therefore, in this study, we analysed kinetic data, including joint power and moment during stance phase, for healthy individuals and preoperative and six-month postoperative patients. Walking speed is a reliable indicator of overall function [22,23]. Furthermore, it is well-established that greater joint moments and power are strongly associated with faster gait speeds, reflecting more efficient and dynamic locomotion [24,25]. However, previous research has shown that individuals often experience a significant decline in walking speed that persists for over a year following surgery [26]. This finding is consistent with other studies reporting long-term gait speed deficits after THA [19,27,28]. Given the critical role of gait speed in functional recovery and its relationship with joint kinetics, this study also examined the effects of walking speed on hip kinetics in individuals post-THA.

Although many studies have explored changes in gait dynamics, there are still deficiencies in gait pattern recognition and gait differences among different sports groups (such as high-low mileage runners) in the current literature [29,30]. Some new methods have been proposed to realize the recognition of human gait patterns, which not only provide a new perspective for sports science, but also provide inspiration for clinical gait analysis. For instance, Xu, Zhou [29] utilized Deep Neural Networks (DNN) and Layer-wise Relevance Propagation (LRP) to analyze gait pattern differences between high- and low-mileage runners, identifying ankle and knee kinematics and kinetics in the sagittal and transverse planes as key discriminators. In addition, another study used machine learning to explain differences in gait patterns between high-mileage and low-mileage runners [30]. However, most of the existing literature focuses on athlete groups or specific gait pattern recognition techniques, and there has been no systematic discussion on the changes of gait dynamics in patients with hip replacement (THA) before and after surgery, especially the specific differences in different subphases of stance phase.

The objectives of this study were threefold: (1) to identify differences in hip moment between participants who have unilateral hip OA after 6 months of THA, (2) to compare them with healthy controls, and (3) to determine the impact of walking speed on hip moment in participants undergoing THA. The same research question was studied in hip power. We hypothesised that the hip moment and power of participants who have unilateral hip OA would increase by 6 months after THA. We also hypothesised that no difference would be identifiable between participants who underwent THA and healthy individuals.

## Methods

A secondary analysis was conducted using a publicly available dataset. For more details regarding this data set, please refer to Bertaux et al. [31].

### Description of the dataset

The dataset used in this study included data from 80 asymptomatic individuals and 106 participants who have unilateral hip OA before THA. Only 92 participants remained to collect gait data six months after the surgery. Table 1 presents the

**Table 1. Demographic data of healthy controls and preoperative and postoperative participants in the dataset.**

| Demographic data Mean(range) | Healthy (n = 80) | Preoperative (n = 106) | Postoperative (n = 92) |
|---|---|---|---|
| Age (years) | 58.7 (25–82) | 66.9 (45–85) | 67.2 (45–85) |
| Sex (M: W) | 35:45 | 51:55 | 44:48 |
| Height (m) | 1.66 (1.49 to 1.87) | 1.64 (1.39 to 1.88) | 1.65 (1.41 to 1.87) |
| Weight (kg) | 69.3 (43.5 to 108.0) | 77.8 (40.0 to 131.0) | 78.3 (44.0 to 136.0) |
| BMI (kg/m²) | 25.0 (17.8 to 33.5) | 28.7 (19.4 to 47.8) | 28.7 (19.3 to 49.4) |

M: men; W: women; BMI: body mass index.

demographic characteristics of the participants. These participants were recruited on a voluntary basis between 2011 and 2016 and the protocol approved by the ethics committee of the administrating institute [31]. It was approved by the local ethic committee [31]. Data on the c3d files of the full-body motion capture trial and corresponding static trial, marker set, joint angles, plug-in gait, bones, centre of mass (CoM), normalised ground reaction force (GRF), and Kellgren–Lawrence grade [31] were included. A diagnosis of hip OA was established in accordance with the American College of Rheumatology criteria [32]. The exclusion criteria were stated in the original study.

The sample size was calculated using G*Power software version 3.1.9.7 (Universities, Düsseldorf, Germany). An effect size (d) of 0.81 was estimated based on data from Foucher et al. [33], which examined differences in hip extension motion before and after surgery. With a power of 80% and an alpha level of 0.05, the minimum sample size required to detect significant differences was determined to be 15. This study included 69 healthy controls and 67 participants who underwent THA. Table 2 summarises the reasons for participant data exclusion. Table 3 presents the demographic characteristics of the participants.

**Table 2. Reasons for participant data exclusion.**

| Reasons for exclusion | Number of participants (Group) | | |
|---|---|---|---|
| | Healthy | Preoperative | Postoperative |
| Missing data regarding body build | 2 | NA | NA |
| Unavailable c3D static trial files | 5 | 9 | 2 |
| Unavailable post-surgery data | NA | 16 | NA |
| Unavailable pre-surgery data | NA | NA | 19 |
| Noisy data | 4 | 14 | 4 |

**Table 3. Demographic data of healthy controls and preoperative participants.**

| Demographic data Mean(range) | Healthy (n = 69) | Preoperative (n = 67) | P-value |
|---|---|---|---|
| Age (years) | 57.7 (25–82) | 67.4 (46–84) | 0.001 |
| Sex (M: W) | 31:38 | 33:34 | 0.613 |
| Height (m) | 1.66 (1.49 to 1.87) | 1.64 (1.39 to 1.88) | 0.132 |
| Weight (kg) | 68.7 (44.0 to 97.0) | 77.3 (50.0 to 131.0) | 0.001 |
| BMI (kg/m²) | 24.7 (17.8 to 33.5) | 28.6 (20.4 to 47.8) | <0.001 |

M: men; W: women; BMI: body mass index

## Data processing

Plug-in gait marker set modelling was performed using Visual3D (v.2023.09.1; C-Motion, Germantown, MD, USA) in accordance with the developer's instructions [34] (Fig 1). Subsequently, the hip moment and power values of the participants were exported. These values were normalised by each participant's body weight.

Kinetic data were filtered using a smoothed second-order low-pass Butterworth filter with a cut-off frequency of 50 Hz. Gait events were identified using the automatic gait event function of Visual3D. The first initial contact and toe-off were identified using a 5-N threshold applied to the vertical GRF data, and the subsequent initial contact was identified using the trajectory of the relevant foot segment. Joint moment and power were time-normalised to represent a percentage of the stance phase (101 timepoints, from initial contact to foot-off for the same foot). To enable a more detailed investigation of changes in gait kinetics during each subphase of the stance phase, the stance phase was manually segmented into four subphases: loading response, mid-stance, terminal stance, and pre-swing. The loading response was defined that one side of foot initially strikes on the ground until the contralateral foot leaves from the ground and normalized to 16 data points, serving as the critical transition period for weight acceptance and shock absorption. The mid-stance phase was the single-leg support period from contralateral toe-off to when the body's center of mass aligns over the stance foot and normalized to 33 data points, representing the body's challenge to maintain stability on one limb. The terminal stance was the period from heel rise to contralateral heel contact and normalized to 34 data points, representing the body's final push-off mechanism. The pre-swing phase was defined that both feet on the ground until one side of foot leaves the ground and normalized to 18 data points, serving as the critical transition from stance to swing [11,35,36]. Separate SPM1D analyses were performed for each subphase, with the results presented in S1–S9 Figs.

## Data analysis

The baseline demographics of the participants were compared using independent $t$ test. The joint moment and power of preoperative and postoperative patients were compared using SPM paired $t$ tests. Differences in joint moment and power between healthy controls and preoperative participants, and between healthy controls and postoperative participants were

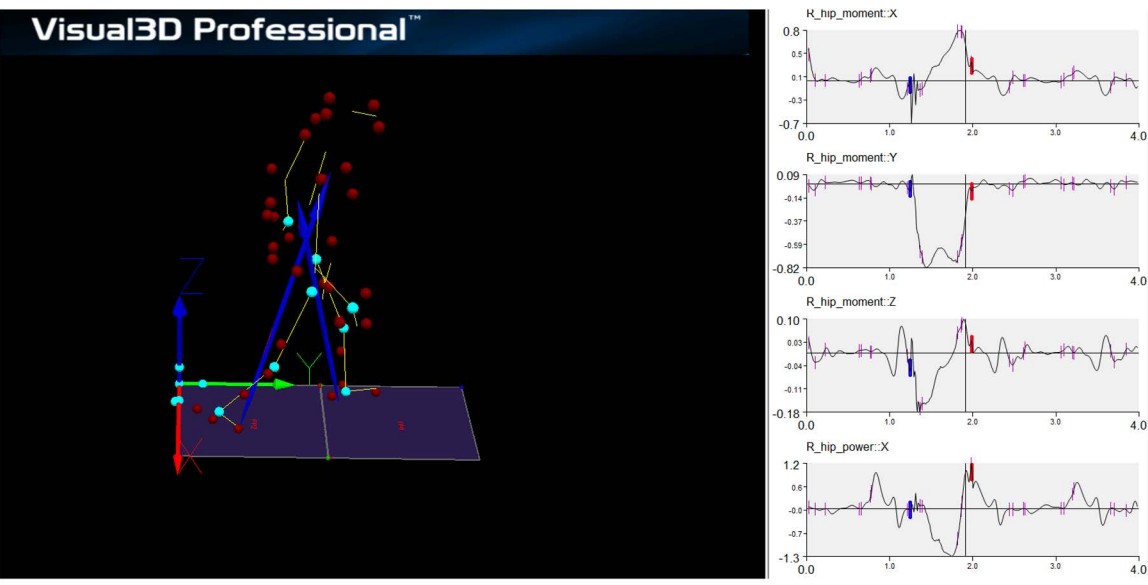

**Fig 1. Illustration of the data processing workflow for gait kinetics using Visual3D.**

examined using SPM independent-samples *t* tests. SPM regression was used to investigate the relationship between changes in walking speed (as the independent variable) and changes in hip moment and power (as the dependent variable) in preoperative and postoperative participants. The SPM1D package does not currently support multiple regression for assessing the predictive power of individual independent variables. Therefore, to determine whether demographic characteristics, including age, sex, and BMI [37,38], are significant predictors, we conducted a multivariable regression analysis using SPSS version 26 (IBM/SPSS Inc., Armonk, NY, USA). This analysis evaluated whether group membership (healthy controls, preoperative participants, or postoperative participants) significantly predicted differences in hip moment and power after adjusting for the effects of the aforementioned covariates. For this analysis, differences in hip moment and power were extracted at the centroids of clusters identified by the SPM1D analysis as showing significant between-group differences. In all scenario, the grouping variable remained a significant predictor. The results of this analysis are presented in S1–S3 Tables. We set a alpha value of 0.05 as a threshold for rejecting the null hypothesis. Comparisons were deemed significant if the test statistics exceeded the rejection threshold at one or more points along the continuum. If the value of SPM1d{t} exceeded the threshold, a cluster was identified, and the corresponding *p* value was calculated and reported. All SPM1d analyses were conducted using the open-source SPM1d package [39] v.0.4.18 in Python v.3.12.2 [40].

## Results

### Hip sagittal moment

Significant differences were observed in the hip sagittal moment between the preoperative and postoperative participants, with a total difference duration of 37.6% in the stance phase (Table 4). The postoperative participants exhibited a higher hip extension moment during the beginning of the loading response phase (0.0–4.3% stance phase, $P = 0.037$). They also exhibited a higher hip flexion moment during the terminal stance and pre-swing phases (66.7–100.0% stance phase, $P < 0.001$; Table 4, Fig 2, S1 Fig). Greater differences in hip sagittal moment at the beginning of the loading response phase (0.3–5.1% stance phase, $P = 0.033$), and the end of the pre-swing phase (95.1–100.0% stance phase, $P = 0.032$) were significantly associated with a higher walking speed (Table 5, Fig 3). During the loading response and terminal stance phases, the hip extension and flexion moments in both preoperative and postoperative participants were below normal levels (Table 4, Fig 2). However, the total duration in the demonstrated significant differences in hip sagittal moment was shorter in postoperative participants compared to preoperative participants.

**Table 4. The SPM1d results of hip sagittal moment.**

| Group | Stance phase cluster time (%) | Subphase | P-value |
|---|---|---|---|
| **Preoperative VS Postoperative** | 0.0-4.3 | Loading response | 0.037[a] |
| | 66.7-100.0 | Terminal-stance, Pre-swing | <0.001[a] |
| | Total: 37.6 | | |
| **Healthy VS Preoperative** | 0.0-4.9 | Loading response | 0.034[b] |
| | 47.6-100.0 | Mid-stance,Terminal-stance, Pre-swing | <0.001[b] |
| | Total: 57.3 | | |
| **Healthy VS Postoperative** | 0.0-2.7 | Loading response | 0.044[b] |
| | 69.8-100.0 | Terminal-stance, Pre-swing | <0.001[b] |
| | Total: 32.9 | | |

[a]P-values were calculated using the SPM paired *t* tests.

[b]P-values were calculated using the SPM independent-samples *t* tests.

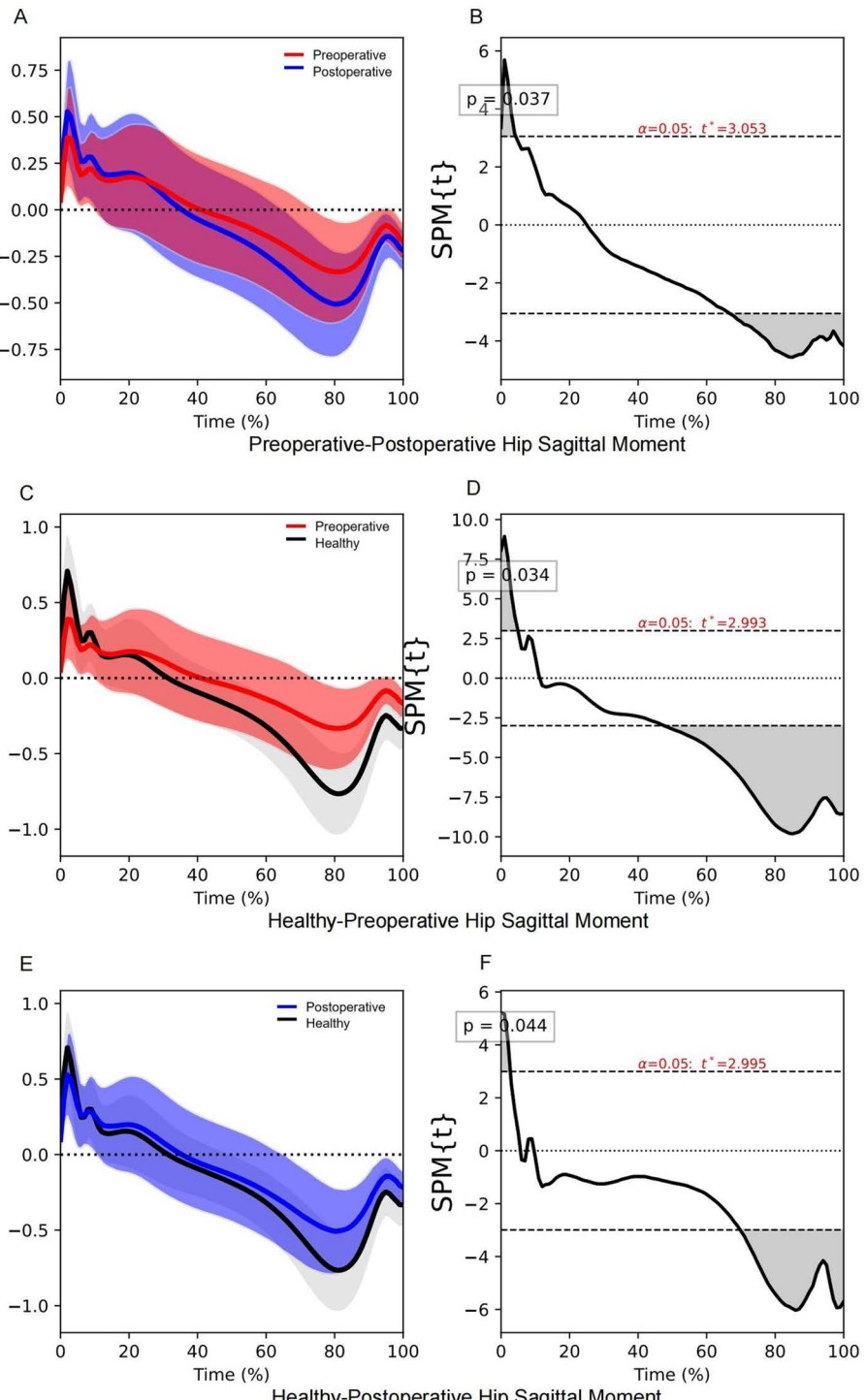

**Fig 2. SPM1d results of hip sagittal moment during the stance phase.** The red line indicates the mean hip sagittal moment of the preoperative participants, the blue line indicates the mean hip sagittal moment of the postoperative participants, and the black line indicates the mean hip sagittal moment of the healthy controls. The shaded areas in panels **(A)**, **(C)**, and **(E)** above and below the line indicate standard deviations, and the grey shaded areas in panels **(B)**, **(D)**, and **(F)** indicate significant differences.

**Table 5. Relationship between changes in walking speed before and after THA and changes in hip moment/power.**

| Kinetic data | Stance phase cluster time (%) | Subphase | P-value |
|---|---|---|---|
| ΔSagittal hip moment | 0.3-5.1 | Loading response | 0.033[c] |
| | 95.1-100.0 | Pre-swing | 0.032[c] |
| | Total: 9.7 | | |
| ΔFrontal hip moment | 7.9-33.5 | Loading response, Mid-stance | <0.001[c] |
| | 82.8-90.0 | Pre-swing | 0.020[c] |
| | Total: 32.8 | | |
| ΔHip power | 1.2-3.1 | Loading response | 0.043[c] |
| | 94.8-100.0 | Pre-swing | 0.017[c] |
| | Total: 7.1 | | |

Δ: Difference before and after THA.

[c]P-values were calculated using the SPM regression.

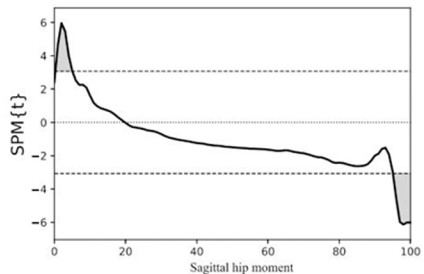 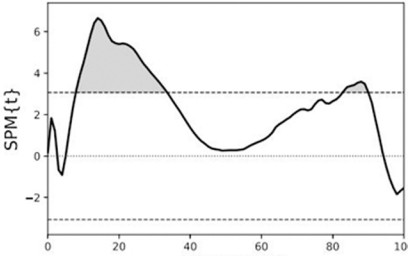 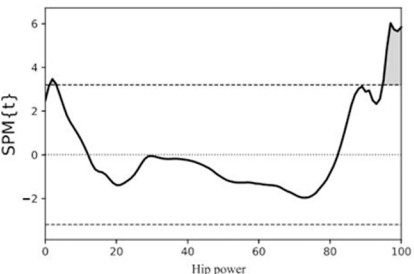

**Fig 3. SPM1d results of relationship between changes in walking speed and in hip power and moment.** The grey shaded areas indicate significant differences.

## Hip frontal moment

Significant differences were observed in the hip frontal moment between the preoperative and postoperative participants, with a total difference duration of 67.3% in the stance phase (Table 6). The hip abduction moment after THA was significantly higher in the postoperative participants during the loading response and mid-stance phases (9.9–38.7% stance phase, $P<0.001$) and during the terminal stance and pre-swing phases (55.2–93.7% stance phase, $P<0.001$; Table 6, Fig 4, S4 Fig). Greater differences in hip frontal moment during the loading response phase, mid-stance phase (7.9–33.5% stance phase, $P<0.001$), and pre-swing phase (82.8–90.0% stance phase, $P=0.020$) were associated with a higher walking speed (Table 5, Fig 3). During the mid-stance and terminal stance, the abduction moment of both preoperative and postoperative participants was below normal (Table 6). However, the total duration in the demonstrated significant differences in hip frontal moment was shorter in postoperative participants compared to preoperative participants.

## Hip power

Significant differences were observed in the hip power between the preoperative and postoperative participants, with a total difference duration of 35.8% in the stance phase (Table 7). In the postoperative participants, the hip joint after THA absorbed more energy during the terminal stance phase (50.8–70.8% stance phase, $P<0.001$) and generated more energy (84.2–100.0% stance phase, $P<0.001$) during the pre-swing phase (Table 7, Fig 5; S7 Fig). Greater differences in hip power during the beginning of the loading response phase (1.2–3.1% stance phase, $P=0.043$), and the end of the

**Table 6. The SPM1d results of hip frontal moment.**

| Group | Stance phase cluster time (%) | Subphase | P-value |
|---|---|---|---|
| **Preoperative VS Postoperative** | 9.9-38.7 | Loading response, Mid-stance | <0.001[a] |
| | 55.2-93.7 | Mid-stance<br>Terminal-stance, Pre-swing | <0.001[a] |
| | Total: 67.3 | | |
| **Healthy VS Preoperative** | 2.7-4.1 | Loading response | 0.048[b] |
| | 8.1-36.0 | Loading response, Mid-stance | <0.001[b] |
| | 63.4-94.0 | Terminal-stance, Pre-swing | <0.001[b] |
| | Total: 59.9 | | |
| **Healthy VS Postoperative** | 2.4-4.1 | Loading response | 0.047[b] |
| | 9.9-28.1 | Loading response, Mid-stance | <0.001[b] |
| | 74.7-92.2 | Terminal-stance, Pre-swing | <0.001[b] |
| | Total: 37.4 | | |

[a]P-values were calculated using the SPM paired $t$ tests.

[b]P-values were calculated using the SPM independent-samples $t$ tests.

pre-swing phase (94.8–100.0% stance phase, $P = 0.017$) were associated with a higher walking speed (Table 5, Fig 3). We also observed that the total duration in stance phase that demonstrated significant differences was shorter in postoperative participants compared to preoperative participants.

## Discussion

In this study, significant differences were observed in multiple types of hip kinetic data between preoperative and postoperative participants who underwent THA. The total length of the SPM1d clusters that exhibited significant differences between healthy controls and the postoperative participants was smaller than that between the healthy controls and the preoperative participants. Despite improvements being observed in the postoperative participants, certain deficits remained in their hip kinetics. In addition, significant differences were observed in hip moment and power before and after THA that were associated with changes in walking speed.

Total hip arthroplasty is commonly used to alleviate pain and improve gait in individuals who have hip OA, leading to better joint stability and mobility [41]. In this study, a significant increase in hip extension moment was observed during the beginning of the loading response phase, which is consistent with the findings of Mont et al. [42]. This suggested THA patients experience improved hip extensor (gluteus maximus and posterior muscles) activation [4] and reduced pain [4,43], allowing for stronger limb force application. Additionally, a significant increase in hip flexion moment was observed during the terminal stance and pre-swing phases, similar to Chopra et al.'s finding [44]. These findings indicate that patients who undergo surgery can effectively prepare for the swing phase of gait because of their improved muscle strength [44], and coordination. Generally, the pre-swing phase plays a key role in generating the momentum required to swing the leg forward, with a higher flexion moment indicating a more forceful and effective gait.

The postoperative participants in this study exhibited a significantly higher hip abduction moment during the loading response, mid-stance, terminal stance, and pre-swing phase, which is consistent with the findings of Mont et al., Foucher et al., and Chopra et al. [3,42,44]. They generated large abduction moments to counteract the reactive moment from the CoM and stabilised the pelvis in the frontal plane, as indicated by Beaulieu et al. [45]. This change in gait pattern is presumably due to the hip abductor muscles contracting more forcefully after surgery [3]. Generally, a higher abduction moment indicates more accurate control and stabilisation of the pelvis, which is essential for a balanced and efficient gait.

During the terminal stance phase, the hip joint of the postoperative patients absorbed more energy, preparing the limb for transitioning from stance to swing phase. This absorption is due to the hip flexors eccentrically controlling hip

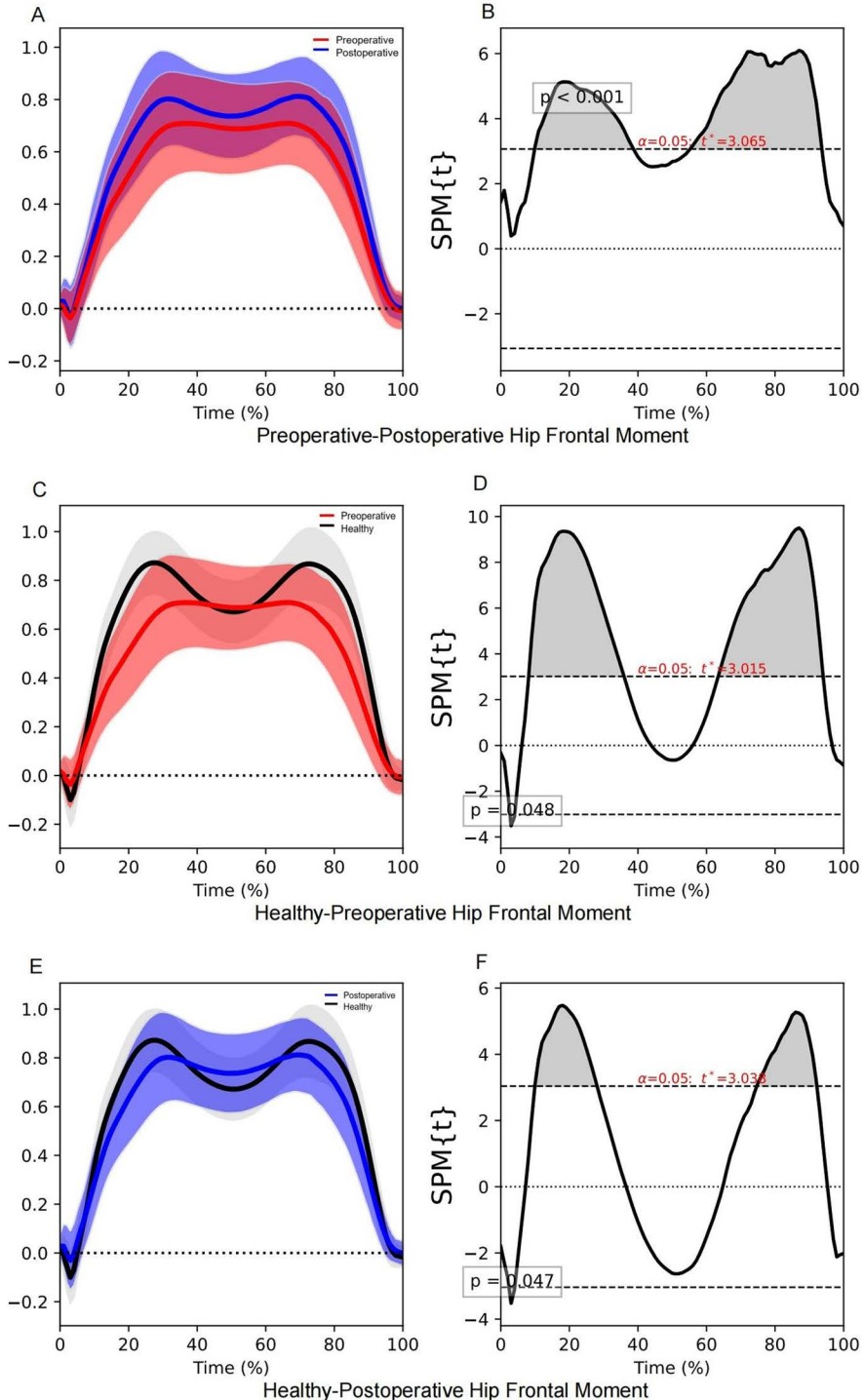

**Fig 4. SPM1d results of hip frontal moment during the stance phase.** The red line indicates the mean hip frontal moment of preoperative participants, the blue line indicates the mean hip frontal moment of postoperative participants, and the black line indicates the mean hip frontal moment of healthy controls. The shaded areas in panels **(A)**, **(C)**, and (E) above and below the line indicate standard deviations, and the grey shaded areas in panels **(B)**, **(D)**, and (F) indicate significant differences.

**Table 7. The SPM1d results of hip power.**

| Group | Stance phase cluster time (%) | Subphase | P-value |
|---|---|---|---|
| **Preoperative VS Postoperative** | 50.8-70.8 | Terminal-stance | <0.001[a] |
| | 84.2-100.0 | Pre-swing | <0.001[a] |
| | Total: 35.8 | | |
| **Healthy VS Preoperative** | 0.0-5.1 | Loading response | 0.024[b] |
| | 10.4-17.4 | Loading response, Mid-stance | 0.013[b] |
| | 44.7-78.4 | Mid-stance, Terminal-stance | <0.001[b] |
| | 82.1-100.0 | Terminal-stance, Pre-swing | <0.001[b] |
| | Total: 63.7 | | |
| **Healthy VS Postoperative** | 10.9-14.1 | Loading response | 0.037[b] |
| | 58.2-74.8 | Terminal-stance | <0.001[b] |
| | 82.7-100.0 | Terminal-stance, Pre-swing | <0.001[b] |
| | Total: 37.1 | | |

[a]P-values were calculated using the SPM paired *t* tests

[b]P-values were calculated using the SPM independent-samples *t* tests

extension, enhanced by better hip ROM [43], muscle function [44], and reduced pain [43] after THA. Such energy absorption plays a key role in stabilising the limb and preparing for the subsequent phase of gait [46]. In this study, before propulsion, high flexion moment was observed, which indicates that the muscles slowed down the limb and prepared for the swing phase, suggesting improved mechanical efficiency in the forces acting on the hip to facilitate movement transitions after THA. During the pre-swing phase, the hip flexes more to initiate leg forward movement, powered by increased concentric hip flexor activity. This absorbed energy aids in advancing the CoM, increasing muscles contraction facilitates lifting and moving the leg forward [44] and enhancing the ability to manage body momentum for a more stable and efficient gait.

We discovered that the hip sagittal and frontal moment and the hip power of the preoperative and postoperative participants were below normal during specific phases. Although this finding contradicts our hypothesis, it is consistent with those of Beaulieu et al. [45], Nantel et al. [47], and Foucher et al. [3]. A systematic review [48] reported postoperative deficits in dynamic lower limb ROM, hip adduction angle, step and stride length, walking speed, and gait symmetry. After THA, patients may continue to adopt long-term pain-avoidance strategies [45], they may attempt to reduce muscle contraction around the hip joint [3], leading to muscle tightness, including hip flexor contracture [49], as well as muscle weakness due to a lack of muscle activation or pain inhibition [3].

Therefore, despite these postoperative improvements, full recovery of gait and joint kinetics may be limited by persistent muscle strength deficits. For instance, Ismailidis et al. found that even 24 months after surgery, hip abductor torque ratios had not returned to normal levels in many patients, suggesting that residual weakness may hinder complete restoration of frontal plane stability [50]. Moreover, Friesenbichler et al. reported that while maximal muscle strength improved after THA, explosive strength remained impaired at 6 months, especially in hip flexors, which are critical for swing-phase initiation [51]. In addition, these patients may continue to exhibit maladaptive gait patterns. These patterns, which develop as a compensatory mechanism during the progression of hip OA before surgery, should be addressed using specific rehabilitation programmes. Resistance training boosts muscle strength, and overall function in postoperative patients [52]. According to our findings, joint strengthening exercises may include a concentric hip flexor strengthening exercise, with the hip in an extended position to facilitate swinging of the lower limb, and a hip abductor strengthening exercise, with translation from hip flexion to a hip extended position to increase the stability of the pelvis. By focusing on these specific

 

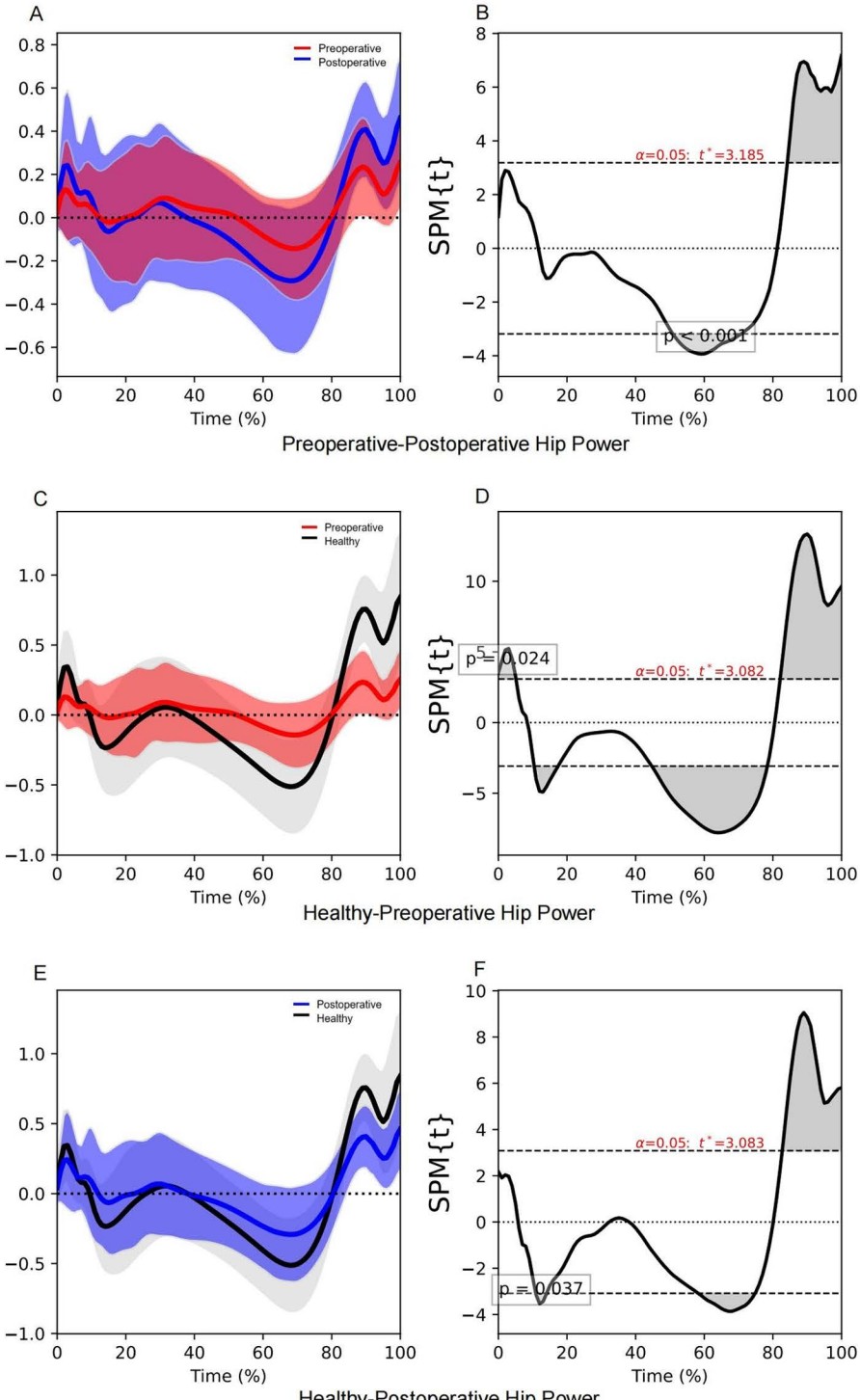

**Fig 5. SPM1d results of hip power during the stance phase.** The red line indicates the mean hip power of the preoperative participants, the blue line indicates the mean hip power of the postoperative participants, and the black line indicates the mean hip power of the healthy controls. The shaded areas in panels **(A)**, **(C)**, and (E) above and below the line indicate standard deviations, and the grey shaded areas in panels **(B)**, **(D)**, and (F) indicate significant differences.

exercises, rehabilitation programmes can effectively address and potentially correct compensatory gait mechanisms and thereby achieve improved mobility and quality of life for patients [53].

The deviations observed from normal kinetics were less pronounced in the postoperative group than in the preoperative group. Similarly, the total length of the SPM1d clusters that exhibited significant differences was smaller in the postoperative group than in the preoperative group, suggesting that THA resulted in a normalisation trend. These findings suggest that our patients, who were examined 6 months after surgery, did not fully recover.

Most of the significant differences associated with changes in walking speed overlapped with those observed in the comparison of hip kinetics in preoperative and postoperative participants. This finding indicates that the increase observed in walking speed partially explains the observed changes. However, this increase in walking speed cannot fully explain the changes observed in kinetics because the region with significant association is relatively small. In addition, we observed no association between changes in walking speed and changes in hip power during the terminal stance phase, for which a between-group difference was observed, indicating other improvements other than changes in walking speed occurred. Further research is required to explore other factors that may have contributed to these improvements.

Our study has several limitations that should be acknowledged. First, we did not report effect sizes because SPM1d does not provide a measure for effect size that would enable comparison of the hip kinetics in patients undergoing THA and healthy individuals. We evaluated differences by determining the total difference in duration between the groups. Second, although pain is an essential covariate, we did not employ any measure of pain levels because the data set that we used did not include information on pain levels. Third, although we observed changes in hip kinetics, other than that for gait speed, we could not determine whether these changes translated into improvements in activities of daily living or quality of life. Therefore, future studies should incorporate functional assessments to provide a more comprehensive evaluation of the outcomes of THA.

## Conclusion

Significant changes are typically observed in hip kinetics during various phases of stance phase after THA, suggesting a trend towards normalisation with improved muscle activation and joint mechanics. Despite some improvements, persistent post-surgery deviations from normal hip mechanics indicate that complete recovery is not achievable within 6 months after surgery. These deviations may be attributable to the maladaptive gait patterns that develop as a compensatory mechanism during the progression of hip OA as well as persistent muscle weakness that does not immediately resolve post-operation. The persistence of these deviations indicates that the path to complete functional recovery is complex and may require extended and targeted rehabilitation that focuses not only on strength and endurance but also on correcting long-term compensatory mechanisms to achieve normal gait mechanics.

## Supporting information

**S1 Fig. SPM paired t test results of hip sagittal moment during the stance phase.** Loading response phase, (B) Mid-stance phase, (C)Terminal stance phase, (D) Pre-swing phase. The grey shaded areas indicate significant differences.
(TIF)

**S2 Fig. SPM independent-samples t test results of hip sagittal moment between healthy controls and preoperative participants.** Loading response phase, (B) Mid-stance phase, (C)Terminal stance phase, (D) Pre-swing phase. The grey shaded areas indicate significant differences.
(TIF)

**S3 Fig. SPM independent-samples t test results of hip sagittal moment between healthy controls and postoperative participants.** Loading response phase, (B) Mid-stance phase, (C)Terminal stance phase, (D) Pre-swing phase. The grey shaded areas indicate significant differences.
(TIF)

**S4 Fig.  SPM paired t test results of hip frontal moment during the stance phase.** Loading response phase, (B) Mid-stance phase, (C)Terminal stance phase, (D) Pre-swing phase. The grey shaded areas indicate significant differences.
(TIF)

**S5 Fig.  SPM independent-samples t test results of hip frontal moment between healthy controls and preoperative participants.** Loading response phase, (B) Mid-stance phase, (C)Terminal stance phase, (D) Pre-swing phase. The grey shaded areas indicate significant differences.
(TIF)

**S6 Fig.  SPM independent-samples t test results of hip frontal moment between healthy controls and postoperative participants.** Loading response phase, (B) Mid-stance phase, (C)Terminal stance phase, (D) Pre-swing phase. The grey shaded areas indicate significant differences.
(TIF)

**S7 Fig.  SPM paired t test results of hip power during the stance phase.** Loading response phase, (B) Mid-stance phase, (C)Terminal stance phase, (D) Pre-swing phase. The grey shaded areas indicate significant differences.
(TIF)

**S8 Fig.  SPM independent-samples t test results of hip power between healthy controls and preoperative participants.** Loading response phase, (B) Mid-stance phase, (C)Terminal stance phase, (D) Pre-swing phase. The grey shaded areas indicate significant differences.
(TIF)

**S9 Fig.  SPM independent-samples t test results of hip power between healthy controls and postoperative participants.** Loading response phase, (B) Mid-stance phase, (C)Terminal stance phase, (D) Pre-swing phase. The grey shaded areas indicate significant differences.
(TIF)

**S1 Table.  The multivariable regression results of hip sagittal moment.**
(DOCX)

**S2 Table.  The multivariable regression results of hip frontal moment.**
(DOCX)

**S3 Table.  The multivariable regression results of hip power.**
(DOCX)

## Author contributions

**Conceptualization:** Lingling Zhong, Patrick Wai-Hang Kwong.

**Formal analysis:** Lingling Zhong, Patrick Wai-Hang Kwong.

**Funding acquisition:** Patrick Wai-Hang Kwong.

**Investigation:** Lingling Zhong, Patrick Wai-Hang Kwong.

**Methodology:** Lingling Zhong, Patrick Wai-Hang Kwong.

**Project administration:** Patrick Wai-Hang Kwong.

**Resources:** Patrick Wai-Hang Kwong.

**Software:** Lingling Zhong, Patrick Wai-Hang Kwong.

**Supervision:** Patrick Wai-Hang Kwong.

**Visualization:** Lingling Zhong, Patrick Wai-Hang Kwong.

**Writing – original draft:** Lingling Zhong.

**Writing – review & editing:** Patrick Wai-Hang Kwong, Jack Jiaqi Zhang, Ananda Sidarta, Clare Chung-Wah Yu.

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
