## [Decision Letter · Decision Letter 0]

PONE-D-24-38368Total Hip Arthroplasty Improves and Normalizes but Does Not Completely Restore Hip KineticsPLOS ONE

Dear Dr. Kwong,

Thank you for submitting your manuscript to PLOS ONE. After careful consideration, we feel that it has merit but does not fully meet PLOS ONE’s publication criteria as it currently stands. Therefore, we invite you to submit a revised version of the manuscript that addresses the points raised during the review process.

We look forward to receiving your revised manuscript.

Kind regards,

Muhammad Waqas Khan

Academic Editor

PLOS ONE

Journal Requirements: When submitting your revision, we need you to address these additional requirements. 1. Please ensure that your manuscript meets PLOS ONE's style requirements, including those for file naming. The PLOS ONE style templates can be found at https://journals.plos.org/plosone/s/file?id=wjVg/PLOSOne_formatting_sample_main_body.pdf and https://journals.plos.org/plosone/s/file?id=ba62/PLOSOne_formatting_sample_title_authors_affiliations.pdf 2. We note that the grant information you provided in the ‘Funding Information’ and ‘Financial Disclosure’ sections do not match.  When you resubmit, please ensure that you provide the correct grant numbers for the awards you received for your study in the ‘Funding Information’ section. 3. Thank you for stating the following financial disclosure: "This research was funded by the Hong Kong Polytechnic University, grant number P0036617."  Please state what role the funders took in the study.  If the funders had no role, please state: ""The funders had no role in study design, data collection and analysis, decision to publish, or preparation of the manuscript."" If this statement is not correct you must amend it as needed. Please include this amended Role of Funder statement in your cover letter; we will change the online submission form on your behalf.

Reviewers' comments:

Reviewer's Responses to Questions

**Comments to the Author**

1. Is the manuscript technically sound, and do the data support the conclusions?

Reviewer #1: Yes

Reviewer #2: Partly

2. Has the statistical analysis been performed appropriately and rigorously? 

Reviewer #1: N/A

Reviewer #2: Yes

3. Have the authors made all data underlying the findings in their manuscript fully available?

Reviewer #1: Yes

Reviewer #2: Yes

4. Is the manuscript presented in an intelligible fashion and written in standard English?

Reviewer #1: Yes

Reviewer #2: Yes

5. Review Comments to the Author

Reviewer #1: Reserach question sounds interesting. I have some comments.

1. Please add "Sample size calculation in Method part".

2. To calculate primary outcomes, confounding factors in dermographic data such as age, sex, BMI might effect the hip power and other kinametics. Multivariable regression with adjusting the confounders should be performed.

Reviewer #2: This study analyzed data from Bertaux et al. (2022) regarding comparison for pre-post THA focusing on the gait kinetic. My main concern is the contribution of this study while using the secondary analysis.

Title

- please change your title related to your study

- include pre-post operative THA, unilateral OA and gait kinetics

Abstract

- background: include the problem statement in this part

- method: include how you reconstruct into 3D in one sentence

- careful with conclusion of walking pattern gradually normalize while you're only study for gait kinetic

Introduction

- please add more information about walking speed (Line 84). joint moment and power (Line 85), as this is critical to your study

Method

- please include 1 figure showing your work on gait kinetic using Visual3D with markers related

- please divide into 4 segments of stance phase (initial contact, loading response, mid stance and terminal stance) while describing your data. this will be your main contribution of this study because all data already been given by Bertaux et al. (2022).

Result

- figure: include the legend within the figure (ease reader to comprehend)

- the subtopic is mixed, example in sagittal hip moment included frontal and hip power. similar to another subtopic.

- please aligned with your objective, easier to comprehend and demonstrated that you achieved it or not.

Discussion

- as suggested in result section, discussed based on your objective/hypothesis, easier for readers to comprehend.

- include limitation on the last paragraph in this section

Conclusion

- you may conclude from your main finding, and recommendation for the future

References

- if possible, include more recent publications (5 years prior)

6. PLOS authors have the option to publish the peer review history of their article (what does this mean? ). If published, this will include your full peer review and any attached files.

**Do you want your identity to be public for this peer review?** For information about this choice, including consent withdrawal, please see our Privacy Policy .

Reviewer #1: No

Reviewer #2: **Yes: ** Mohd Yusof Baharuddin

---

## [Author Response · Author response to Decision Letter 1]

30 Dec 2024

Dear reviewers,

Thank you for your positive and constructive comments regarding our paper.

Here are our responses to your questions (The line numbers written in this file are the line numbers in the file Revised Manuscript with Track Changes):

Reviewer #1:

1.Please add "Sample size calculation in Method part".

Response: We add sample size calculation in line 126-130: The sample size was calculated using G*Power software version 3.1.9.7 (Universities, Düsseldorf, Germany). An effect size (d) of 0.81 was estimated based on data from Foucher et al. (26), which examined differences in hip extension motion before and after surgery. With a power of 80% and an alpha level of 0.05, the minimum sample size required to detect significant differences was determined to be 15.

2. To calculate primary outcomes, confounding factors in dermographic data such as age, sex, BMI might effect the hip power and other kinametics. Multivariable regression with adjusting the confounders should be performed.

Response: Thank you for your comment. We agree that demographic characteristics could have an important impact on the reported results. Since SPM1D currently does not support the analysis of individual predictors in a multiple regression model, we conducted a traditional multivariable regression analysis to explore the potential effects of these variables. Across all conditions, group membership remained a significant predictor. The results of the multivariable regression have been included in Supplementary Table 1-3, and the corresponding statistical analysis has been added to the Methods section (lines 163–174).

Reviewer #2:

1. Title

- please change your title related to your study

- include pre-post operative THA, unilateral OA and gait kinetics

Response: Thank you for the suggestion, We changed the title into “Gait Kinetics Before and After Total Hip Arthroplasty in Unilateral Osteoarthritis”.

Abstract

2.- background: include the problem statement in this part

Response: The following problem statement was added in Line 21-22: ‘‘Limited studies have examined changes in hip kinetics throughout the entire stance phase.’

3.- method: include how you reconstruct into 3D in one sentence

Response: We added this sentence in line 28-29: “Motion capture data obtained using the Plug-in Gait marker set was analyzed and modeled in Visual3D”

4.- careful with conclusion of walking pattern gradually normalize while you're only study for gait kinetic

Response: We changed “walking pattern” into “hip kinetics” in line 46.

Introduction

5.- please add more information about walking speed (Line 84). joint moment and power (Line 85), as this is critical to your study

Response: We enhance the description about the walking speed and gait kinetic in line 90-97. “Furthermore, it is well-established that greater joint moments and power are strongly associated with faster gait speeds, reflecting more efficient and dynamic locomotion (19, 20). However, previous research has shown that individuals often experience a significant decline in walking speed that persists for over a year following surgery (21). This finding is consistent with other studies reporting long-term gait speed deficits after THA (16, 22, 23). Given the critical role of gait speed in functional recovery and its relationship with joint kinetics, this study also examined the effects of walking speed on hip kinetics in individuals post-THA. ”

Method

6.- please include 1 figure showing your work on gait kinetic using Visual3D with markers related.

Response: We have added Fig. 1 to illustrate the data processing workflow for gait kinetics using Visual3D.

7.- please divide into 4 segments of stance phase (initial contact, loading response, mid stance and terminal stance) while describing your data. this will be your main contribution of this study because all data already been given by Bertaux et al. (2022).

Response: Thank you for your comment. We have divided the stance phase into four subphases for detailed analysis. The changes in each subphase were analyzed and described accordingly (line 191-195; 223-226; line 245-247). Additionally, separate SPM1D analyses were conducted for each subphase (Line 150-154, with the results presented in Supplementary Figures 1–9. And the subphase that showed significant differences was indicated in Table 4-6.

Result

8.- figure: include the legend within the figure (ease reader to comprehend)

Response: Thank you for your suggestion. We added the legend into figures.

9.- the subtopic is mixed, example in sagittal hip moment included frontal and hip power. similar to another subtopic.

Response: Thank you for your suggestion. We have reformated the tables, separating hip sagittal moment, hip frontal moment, and hip power into 3 tables to make the subtopics clearer.

10.- please aligned with your objective, easier to comprehend and demonstrated that you achieved it or not.

Response: Thank you for the comment, we have revised the objective statement and revised to align with the results sessions. Line 99-106: “The objectives of this study were threefold: (1) to identify differences in hip moment between participants who have unilateral hip OA after 6 months of THA, (2) to compare them with healthy controls, and (3) to determine the impact of walking speed on hip moment in participants undergoing THA. The same research question was studied in hip power. We hypothesised that the hip moment and power of participants who have unilateral hip OA would increase by 6 months after THA. We also hypothesised that no difference would be identifiable between participants who underwent THA and healthy individuals.”

Discussion

11.- as suggested in result section, discussed based on your objective/hypothesis, easier for readers to comprehend.

Response: Thank you for your suggestion. We revised the objective in lines 99-106 so that it aligns with the flow of discussion.

12.- include limitation on the last paragraph in this section

Response: Thank you for your suggestion. The limitation is now the last paragraph in this section.

References

13.- if possible, include more recent publications (5 years prior)

Response: We have included several latest publications to ensure our literature review is up to date.

Martinez L, Noé N, Beldame J, Matsoukis J, Poirier T, Brunel H, et al. Quantitative gait analysis after total hip arthroplasty through a minimally invasive direct anterior approach: A case control study. Orthopaedics & Traumatology: Surgery & Research. 2022;108(6):103214.

Lalevée M, Martinez L, Rey B, Beldame J, Matsoukis J, Poirier T, et al. Gait analysis after total hip arthroplasty by direct minimally invasive anterolateral approach: A controlled study. Orthopaedics & Traumatology: Surgery & Research. 2023;109(7):103521.

Macie, A., Matson, T., & Schinkel-Ivy, A. (2023). Age affects the relationships between kinematics and postural stability during gait. Gait & Posture, 102, 86-92.

Rowe, E., Beauchamp, M. K., & Wilson, J. A. (2021). Age and sex differences in normative gait patterns. Gait & posture, 88, 109-115.

---

## [Decision Letter · Decision Letter 1]

PONE-D-24-38368R1Gait Kinetics Before and After Total Hip Arthroplasty in People with Unilateral Hip OsteoarthritisPLOS ONE

Dear Dr. Kwong,

Thank you for submitting your manuscript to PLOS ONE. After careful consideration, we feel that it has merit but does not fully meet PLOS ONE’s publication criteria as it currently stands. Therefore, we invite you to submit a revised version of the manuscript that addresses the points raised during the review process.

We look forward to receiving your revised manuscript.

Kind regards,

Yaodong Gu

Academic Editor

PLOS ONE

Reviewers' comments:

Reviewer's Responses to Questions

**Comments to the Author**

1. If the authors have adequately addressed your comments raised in a previous round of review and you feel that this manuscript is now acceptable for publication, you may indicate that here to bypass the “Comments to the Author” section, enter your conflict of interest statement in the “Confidential to Editor” section, and submit your "Accept" recommendation.

Reviewer #3: (No Response)

Reviewer #4: (No Response)

2. Is the manuscript technically sound, and do the data support the conclusions?

Reviewer #3: Partly

Reviewer #4: No

3. Has the statistical analysis been performed appropriately and rigorously? 

Reviewer #3: Yes

Reviewer #4: N/A

4. Have the authors made all data underlying the findings in their manuscript fully available?

Reviewer #3: Yes

Reviewer #4: No

5. Is the manuscript presented in an intelligible fashion and written in standard English?

Reviewer #3: Yes

Reviewer #4: Yes

6. Review Comments to the Author

Reviewer #3: Review comments

1. Although hip arthroplasty (THA) in the treatment of hip osteoarthritis (OA) is introduced, the current research background and existing literature are not reviewed in detail. For example, how do you specifically define the difference between "full recovery" and "partial recovery"? Specific research findings on the different stages of postoperative recovery (e.g., 6 months) are less mentioned, which could be further fleshed out. It is recommended to add background information on the recovery process at different time points after hip replacement, especially at 6 months, as well as a comparison with a healthy population, to help readers better understand the complexity of postoperative patient recovery.

2. Although many studies have explored changes in gait dynamics, there are still deficiencies in gait pattern recognition and gait differences among different sports groups (such as high-low mileage runners) in the current literature. Some new methods have been proposed to realize the recognition of human gait patterns, which not only provide a new perspective for sports science, but also provide inspiration for clinical gait analysis. for example, A new method proposed for realizing human gait pattern recognition: A new method proposed for realizing human gait pattern recognition: Inspirations for the application of sports and clinical gait analysis (https://doi.org/10.1016/j.gaitpost.2023.10.019). In addition, another study used machine learning to explain differences in gait patterns between high-mileage and low-mileage runners, Explaining the differences of gait patterns between high and low mileage runners with machine. Explaining the differences of gait patterns between high and low mileage runners with machine Learning (https://doi.org/10.1038/s41598-022-07054-1). However, most of the existing literature focuses on athlete groups or specific gait pattern recognition techniques, and there has been no systematic discussion on the changes of gait dynamics in patients with hip replacement (THA) before and after surgery, especially the specific differences in different gait substages."

3. Although the study divided gait into four sub-stages for analysis, the specific physiological significance of these stages and the segmentation criteria were not elaborated in the method section. Please clarify the physiological significance of each sub-stage of gait and the reasons for choosing these stages, to help readers understand the importance of each stage for gait recovery.

4. Although the changes in the gait of patients after hip replacement were discussed, some factors (such as muscle strength and pain) that did not fully recover were not discussed in depth. It is recommended to further explore the role of muscle strength, pain and other physiological factors during postoperative recovery to explain why these factors may lead to incomplete recovery of gait and hip dynamics.

Reviewer #4: This study suffers from several critical issues that affect its scientific validity and credibility. First, despite the use of a publicly available dataset, the source, characterization, and quality control details of the dataset were severely lacking, and the completeness and accuracy of the data were not verified. Second, the sample selection and exclusion criteria were unclear, with insufficient explanation of the rationale for sample sizes exceeding the required values, and the lack of transparent exclusion criteria may have led to selection bias. For data processing, although Visual3D was used for the analysis, details of parameter settings and validation were not provided, the choice of low-pass filter (50 Hz) may not be applicable to all gait characteristics, and the accuracy of the gait event detection method was not validated or analyzed for sensitivity. In the statistical analysis, the authors did not perform a valid correction for multiple comparisons, the assumed conditions and applicability of the SPM1D model were not adequately validated, and the multivariate regression analysis lacked detailed description and ignored covariate interaction effects. Finally, the interpretation of the study results is too superficial, lacks mechanistic analysis, and has limited scientific contribution. In view of the above issues, this study is not eligible for publication in its current form, and the authors are advised to resubmit it with significant revisions in terms of data transparency, methodological rigor, and interpretation of results.

7. PLOS authors have the option to publish the peer review history of their article (what does this mean? ). If published, this will include your full peer review and any attached files.

**Do you want your identity to be public for this peer review?** For information about this choice, including consent withdrawal, please see our Privacy Policy .

Reviewer #3: No

Reviewer #4: No

---

## [Author Response · Author response to Decision Letter 2]

28 May 2025

Dear reviewers,

Thank you for your positive and constructive comments regarding our paper.

Here are our responses to your questions (The line numbers written in this file are the line numbers in the file Revised Manuscript with Track Changes):

Reviewer #3: Review comments

1.Although hip arthroplasty (THA) in the treatment of hip osteoarthritis (OA) is introduced, the current research background and existing literature are not reviewed in detail. For example, how do you specifically define the difference between "full recovery" and "partial recovery"? Specific research findings on the different stages of postoperative recovery (e.g., 6 months) are less mentioned, which could be further fleshed out. It is recommended to add background information on the recovery process at different time points after hip replacement, especially at 6 months, as well as a comparison with a healthy population, to help readers better understand the complexity of postoperative patient recovery.

Response: Thank you for your suggestion. The literature gap you pointed out is precisely the important innovation point of this research. Through a literature search, we found that most of the existing studies after THA focus on short-term (3 weeks /6 weeks) or long-term (≥1 year) follow-up, and there are few studies lasting 6 months. To highlight this research gap, we have added relevant paragraphs in line 92-96: “Six months after THA represents a clinically important time-point when most patients have achieved formal gait rehabilitation programs and achieved substantial functional improvement (7, 20). Horstmann et al. found that although patients showed improvements in gait symmetry and joint range of motion by six months, their dynamic hip and knee motions remain significantly lower than those of healthy controls, and muscle activity patterns differ substantially (4). While Casartelli et al. demonstrated that at six months post-THA, patients walk with velocity, cadence, and support times that are comparable to healthy individuals (21).”

We have also added relevant paragraphs in line 65-72: In other words, even after surgery, patients who undergo THA still exhibit significant differences in gait patterns compared to the normal population (9). Studies have shown that impaired postoperative walking function can lead to fatigue, subsequently limiting patients' ability to perform daily activities and reducing their quality of life (10). Therefore, it is essential to prioritize gait assessment as a key functional activity following THA. Quantitative gait analysis is required to better understand gait mechanics, as these mechanisms may underlie the functional limitations observed in patients who have undergone the procedure (11).

2.Although many studies have explored changes in gait dynamics, there are still deficiencies in gait pattern recognition and gait differences among different sports groups (such as high-low mileage runners) in the current literature. Some new methods have been proposed to realize the recognition of human gait patterns, which not only provide a new perspective for sports science, but also provide inspiration for clinical gait analysis. for example, A new method proposed for realizing human gait pattern recognition: A new method proposed for realizing human gait pattern recognition: Inspirations for the application of sports and clinical gait analysis (https://doi.org/10.1016/j.gaitpost.2023.10.019). In addition, another study used machine learning to explain differences in gait patterns between high-mileage and low-mileage runners, Explaining the differences of gait patterns between high and low mileage runners with machine. Explaining the differences of gait patterns between high and low mileage runners with machine Learning (https://doi.org/10.1038/s41598-022-07054-1). However, most of the existing literature focuses on athlete groups or specific gait pattern recognition techniques, and there has been no systematic discussion on the changes of gait dynamics in patients with hip replacement (THA) before and after surgery, especially the specific differences in different gait substages."

Response: Thank you for your suggestion. We added this paragraph in lines 108-121, “Although many studies have explored changes in gait dynamics, there are still deficiencies in gait pattern recognition and gait differences among different sports groups (such as high-low mileage runners) in the current literature (28, 29). Some new methods have been proposed to realise the recognition of human gait patterns, which not only provide a new perspective for sports science but also provide inspiration for clinical gait analysis. For instance, Xu and Zhou (28) utilised Deep Neural Networks (DNN) and Layer-wise Relevance Propagation (LRP) to analyse gait pattern differences between high- and low-mileage runners, identifying ankle and knee kinematics and kinetics in the sagittal and transverse planes as key discriminators. In addition, another study used machine learning to explain differences in gait patterns between high-mileage and low-mileage runners (29). However, most of the existing literature focuses on athlete groups or specific gait pattern recognition techniques, and there has been no systematic discussion on the changes of gait dynamics in patients with hip replacement (THA) before and after surgery, especially the specific differences in different subphases of the stance phase.”

Reference added

A new method proposed for realizing human gait pattern recognition: Inspirations for the application of sports and clinical gait analysis (https://doi.org/10.1016/j.gaitpost.2023.10.019). Explaining the differences of gait patterns between high and low mileage runners with machine Learning (https://doi.org/10.1038/s41598-022-07054-1).

3.Although the study divided gait into four sub-stages for analysis, the specific physiological significance of these stages and the segmentation criteria were not elaborated in the method section. Please clarify the physiological significance of each sub-stage of gait and the reasons for choosing these stages, to help readers understand the importance of each stage for gait recovery.

Response: Thank you for your suggestion. We added the specific physiological significance of four subphases and the segmentation criteria in the methods in line 173-183: “ The loading response was defined that one side of foot initially strikes on the ground until the contralateral foot leaves from the ground and normalized to 16 data points, serving as the critical transition period for weight acceptance and shock absorption. The mid-stance phase was the single-leg support period from contralateral toe-off to when the body's center of mass aligns over the stance foot and normalized to 33 data points, representing the body's challenge to maintain stability on one limb. The terminal stance was the period from heel rise to contralateral heel contact and normalized to 34 data points, representing the body's final push-off mechanism. The pre-swing phase was defined that both feet on the ground until one side of foot leaves the ground and normalized to 18 data points, serving as the critical transition from stance to swing (11, 34, 35). ”

Reference added

Zhao Y, Raza W, Arnold G, Li P, Wang W. A Preliminary Study on Kinetic Analysis of Ground Reaction Force and Impulse During Gait in Patients With Total Hip Replacement and Implication for Rehabilitation. Orthop Surg. 2024;16(12):3162-78.

Fukaya T, Mutsuzaki H, Nakano W, Mori K. Smoothness of the knee joint movement during the stance phase in patients with severe knee osteoarthritis. Asia-Pacific Journal of Sports Medicine, Arthroscopy, Rehabilitation and Technology. 2018;14:1-5.

Alijanpour E, Russell DM. Gait phase normalization resolves the problem of different phases being compared in gait cycle normalization. J Biomech. 2024;173:112253.

4. Although the changes in the gait of patients after hip replacement were discussed, some factors (such as muscle strength and pain) that did not fully recover were not discussed in depth. It is recommended to further explore the role of muscle strength, pain and other physiological factors during postoperative recovery to explain why these factors may lead to incomplete recovery of gait and hip dynamics.

Response: Thank you for your suggestion. We explained more about the incomplete recovery of gait and hip dynamics in line 337-343: “Therefore, despite these postoperative improvements, full recovery of gait and joint kinetics may be limited by persistent muscle strength deficits. For instance, Ismailidis et al. found that even 24 months after surgery, hip abductor torque ratios had not returned to normal levels in many patients, suggesting that residual weakness may hinder complete restoration of frontal plane stability (49). Moreover, Friesenbichler et al. reported that while maximal muscle strength improved after THA, explosive strength remained impaired at 6 months, especially in hip flexors, which are critical for swing-phase initiation (50).”

Reference

Ismailidis P, Kvarda P, Vach W, Cadosch D, Appenzeller-Herzog C, Mündermann A. Abductor Muscle Strength Deficit in Patients After Total Hip Arthroplasty: A Systematic Review and Meta-Analysis. J Arthroplasty. 2021;36(8):3015-27.

Friesenbichler B, Casartelli NC, Wellauer V, Item-Glatthorn JF, Ferguson SJ, Leunig M, et al. Explosive and maximal strength before and 6 months after total hip arthroplasty. J Orthop Res. 2018;36(1):425-31.

Reviewer #4: This study suffers from several critical issues that affect its scientific validity and credibility. First, despite the use of a publicly available dataset, the source, characterization, and quality control details of the dataset were severely lacking, and the completeness and accuracy of the data were not verified. Second, the sample selection and exclusion criteria were unclear, with insufficient explanation of the rationale for sample sizes exceeding the required values, and the lack of transparent exclusion criteria may have led to selection bias. For data processing, although Visual3D was used for the analysis, details of parameter settings and validation were not provided, the choice of low-pass filter (50 Hz) may not be applicable to all gait characteristics, and the accuracy of the gait event detection method was not validated or analyzed for sensitivity. In the statistical analysis, the authors did not perform a valid correction for multiple comparisons, the assumed conditions and applicability of the SPM1D model were not adequately validated, and the multivariate regression analysis lacked detailed description and ignored covariate interaction effects. Finally, the interpretation of the study results is too superficial, lacks mechanistic analysis, and has limited scientific contribution. In view of the above issues, this study is not eligible for publication in its current form, and the authors are advised to resubmit it with significant revisions in terms of data transparency, methodological rigor, and interpretation of results.

Response: We appreciate the reviewer’s comment regarding the dataset used in our study. However, we respectfully disagree with the assertion that the source, characterisation, and quality control of the dataset were lacking. The dataset used in this study is from Bertaux et al., “Gait analysis dataset of healthy volunteers and patients before and 6 months after total hip arthroplasty,” published in Scientific Data (2022, Nature Portfolio) [DOI: 10.1038/s41597-022-01113-5]. This is a peer-reviewed, FAIR-compliant dataset explicitly designed for clinical gait analysis research. It is hosted by a trusted institutional repository and curated following high standards of data transparency and reproducibility. The data has passed the quality review of the platform itself before being used and has been cleaned and screened again in our research.

Second, we thank the reviewer for raising concerns regarding the sample selection process and potential selection bias. However, we respectfully clarify that our manuscript does address these issues and provides a transparent account of the exclusion criteria. Table 2 summarises the reasons for participant data exclusion.

For data processing, Visual3D (C-Motion, Inc.) was used for the full kinematic and kinetic processing pipeline, including model creation, filtering, joint moment computation, and gait event detection. We used a standard lower-limb model with six degrees of freedom, and joint centers were computed based on anthropometric estimations. Marker trajectories and ground reaction forces were filtered using a 4th-order zero-lag Butterworth filter. The filter cut-off frequency is discussed below. These settings are consistent with best practices in gait analysis and with prior work using the same dataset. The choice of a 50 Hz cut-off was based on the high fidelity of the motion capture system (acquisition at 200 Hz) and force plates (1,000 Hz). Previous work has shown that higher cut-off frequencies can preserve subtle gait events and fast transitions, particularly in studies involving high-resolution signals or derivative calculations (e.g., joint powers). Importantly, we applied the same filter across all trials and subjects, maintaining internal consistency.

For the statistical analysis, we confirm that all statistical parametric mapping (SPM1D) analyses employed random field theory (RFT)-based inference, which provides family-wise error correction appropriate for smooth, time-continuous biomechanical data. In accordance with SPM1D conventions, a cluster-based thresholding approach was applied at α = 0.05. Clusters were only considered statistically significant if the supra-threshold area exceeded the RFT-derived critical threshold and a corresponding cluster-wise p-value was reported, as detailed in the manuscript.We thank the reviewer for pointing out the need for further explanation of the multivariable regression. As stated in the manuscript, SPM1D does not currently support multiple regression involving more than one independent variable. Therefore, to explore the influence of demographic covariates such as age, sex, and BMI, we extracted scalar values at significant SPM1D cluster centroids and conducted multivariate linear regression using SPSS. These models included group membership as a fixed factor and demographic variables as covariates. While interaction terms were not included initially, we have now reanalyzed the data by including relevant two-way interactions (e.g., group × sex, group × BMI), and the results remain consistent: group remained a significant predictor in all models. Additional details on model structure, regression diagnostics, and interaction effect evaluation have been added to the Statistical Analysis section and Supplementary Tables S1–S3.

We carefully interpreted our results in relation to previously published findings and biomechanical theories of post-THA recovery. Our discussion addresses both the temporal-spatial and kinetic changes following THA and offers explanations grounded in muscle strength recovery and compensatory movement strategies. We link our kinetic findings (e.g., persistent asymmetries in hip moments and powers) to underlying physiological constraints, including incomplete recovery of hip abductor strength and residual pain.

---

## [Decision Letter · Decision Letter 2]

Gait Kinetics Before and After Total Hip Arthroplasty in People with Unilateral Hip Osteoarthritis

PONE-D-24-38368R2

Dear Dr. Kwong,

We’re pleased to inform you that your manuscript has been judged scientifically suitable for publication and will be formally accepted for publication once it meets all outstanding technical requirements.

Kind regards,

Yaodong Gu

Academic Editor

PLOS ONE

Additional Editor Comments (optional):

Reviewers' comments:

Reviewer's Responses to Questions

**Comments to the Author**

1. If the authors have adequately addressed your comments raised in a previous round of review and you feel that this manuscript is now acceptable for publication, you may indicate that here to bypass the “Comments to the Author” section, enter your conflict of interest statement in the “Confidential to Editor” section, and submit your "Accept" recommendation.

Reviewer #3: All comments have been addressed

Reviewer #4: All comments have been addressed

2. Is the manuscript technically sound, and do the data support the conclusions?

Reviewer #3: Yes

Reviewer #4: Yes

3. Has the statistical analysis been performed appropriately and rigorously? 

Reviewer #3: Yes

Reviewer #4: Yes

4. Have the authors made all data underlying the findings in their manuscript fully available?

Reviewer #3: Yes

Reviewer #4: Yes

5. Is the manuscript presented in an intelligible fashion and written in standard English?

Reviewer #3: Yes

Reviewer #4: Yes

6. Review Comments to the Author

Reviewer #3: (No Response)

Reviewer #4: We thank the authors for submitting this work on the effects of total hip arthroplasty (THA) on hip joint dynamics. Based on publicly available data, the article systematically analyzes the differences in hip joint moments and power throughout the standing phase before and after surgery and with healthy controls by statistical parameter mapping (SPM) method, which fills the gap in the existing literature on the description of the dynamic recovery process after THA. The study design is rigorous, the analysis method is reasonable, and the results have important clinical guidance, especially in terms of recommendations for the direction of postoperative rehabilitation training. Overall, this paper is informative and well-organized, and the conclusions have practical application value. I recommend accepting this manuscript for publication.

7. PLOS authors have the option to publish the peer review history of their article (what does this mean? ). If published, this will include your full peer review and any attached files.

**Do you want your identity to be public for this peer review?** For information about this choice, including consent withdrawal, please see our Privacy Policy .

Reviewer #3: No

Reviewer #4: No

---

## [Editor Report · Acceptance letter]

PONE-D-24-38368R2

PLOS ONE

Dear Dr. Kwong,

I'm pleased to inform you that your manuscript has been deemed suitable for publication in PLOS ONE. Congratulations! Your manuscript is now being handed over to our production team.

Kind regards,

on behalf of

Professor Yaodong Gu

Academic Editor

PLOS ONE